# HeNCler: Node Clustering in Heterophilous Graphs via Learned Asymmetric Similarity

## Abstract

Clustering nodes in heterophilous graphs is challenging as traditional methods assume that effective clustering is characterized by high intra-cluster and low inter-cluster connectivity. To address this, we introduce HeNCler —a novel approach for **He**terophilous **N**ode **Clu**ster**ing. HeNCler *learns* a similarity graph by optimizing a clustering-specific objective based on weighted kernel singular value decomposition. Our approach enables spectral clustering on an *asymmetric* similarity graph, providing flexibility for both directed and undirected graphs. By solving the primal problem directly, our method overcomes the computational difficulties of traditional adjacency partitioning-based approaches. Experimental results show that HeNCler significantly improves node clustering performance in heterophilous graph settings, highlighting the advantage of its asymmetric graph-learning framework.

## 1 Introduction

Graph neural networks (GNNs) have substantially advanced machine learning applications to graph-structured data by effectively propagating node attributes end-to-end. Typically, GNNs rely on the assumption of homophily, where nodes with similar labels are more likely to be connected (Zheng et al., 2024; Wu et al., 2021). The homophily assumption holds true in contexts such as social networks and citation graphs, where models like GCN (Kipf & Welling, 2017), GIN (Xu et al., 2019), and GraphSAGE (Hamilton et al., 2017) excel at tasks like node classification and graph prediction.

However, in heterophilous datasets, such as web page and transaction networks, edges often link nodes with differing labels. Models like GAT (Veličković et al., 2018) and various graph transformers (Ying et al., 2022; Dwivedi & Bresson, 2021) have demonstrated improved performance on these datasets. Their attention mechanisms learning edge importance provide a straightforward way to reduce the reliance on homophily for supervised tasks.

Our work specifically addresses unsupervised attributed node clustering tasks. Such tasks necessitate entirely unsupervised or self-supervised learning approaches. For instance, auto-encoder type models (Park et al., 2019; Pan et al., 2020) are primarily focused on node representation learning rather than clustering, making them less suited for directly improving cluster-ability. Various self-supervised, contrastive learning techniques (Hassani & Ahmadi, 2020; You et al., 2020) enhance node representation learning in homophilous settings only and lack a specific clustering objective. At the same time, several self-supervised methods have been developed to handle heterophilous graphs (Chen et al., 2022; Xiao et al., 2022; Yuan et al., 2023). For example, MUSE (Yuan et al., 2023) extracts semantic and contextual views for contrastive learning. However, these methods are designed for the general node representation learning task and lack a clustering objective.

In contrast, S³GC (Devvrit et al., 2022) employs a self-supervised approach specifically designed for clustering. It however assumes homophily by leveraging random walk co-occurrences to infer proximity-based similarities. MinCutPool (Bianchi et al., 2020) and DMoN (Tsitsulin et al., 2023) introduce unsupervised losses linked to graph structure, with strong theoretical ties to spectral clustering and graph modularity, respectively. These methods are suited for undirected graphs only, and moreover rely on partitioning the adjacency matrix where effective clustering correlates with high intra-cluster and low inter-cluster similarity—a premise often invalid in heterophilous graphs.

This paper introduces HeNCler, a novel approach for node clustering in heterophilous graphs, illustrated in Figure 1. Existing works overlook the asymmetric relationships in heterophilous

Table 1: Qualitative comparison of HeNCler with several baselines. In the table, $|\mathcal{V}|$, $|\mathcal{B}|$, and $|\mathcal{E}|$ denote the total number of nodes, the mini-batch size, and the number of edges respectively.

| | BASELINES | | | | OURS |
| --- | --- | --- | --- | --- | --- |
| | MINCUTP. | DMON | S³GC | MUSE | HENCLER |
| CAN HANDLE HETEROPHILY | ✗ | ✗ | ✗ | ✓ | ✓ |
| DIRECTED GRAPHS | ✗ | ✗ | ✓ | ✓ | ✓ |
| HAS CLUSTERING OBJECTIVE | ✓ | ✓ | ✓ | ✗ | ✓ |
| SPACE COMPLEXITY | $\mathcal{O}(|\mathcal{V}|^2)$ | $\mathcal{O}(|\mathcal{V}| + |\mathcal{E}|)$ | $\mathcal{O}(|\mathcal{B}|)$ | $\mathcal{O}(|\mathcal{V}| + |\mathcal{E}|)$ | $\mathcal{O}(|\mathcal{B}|)$ |
| TIME COMPLEXITY | $\mathcal{O}(|\mathcal{V}| + |\mathcal{E}|)$ | $\mathcal{O}(|\mathcal{V}| + |\mathcal{E}|)$ | $\mathcal{O}(|\mathcal{V}|)$ | $\mathcal{O}(|\mathcal{V}| + |\mathcal{E}|)$ | $\mathcal{O}(|\mathcal{V}|)$ |

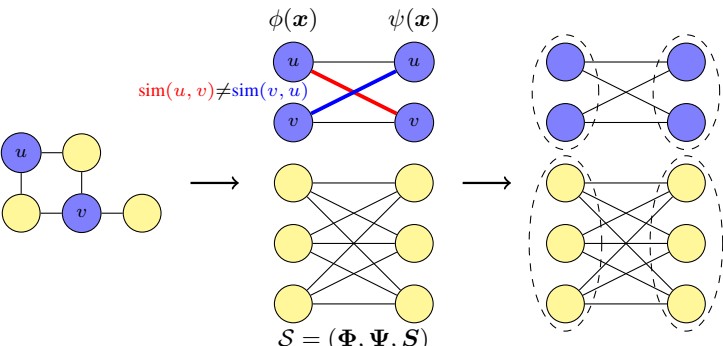

Figure 1: **HeNCler Overview**. Starting from a heterophilous graph, where nodes with the same label are not close to each other (left), HeNCler learns two sets of node representations, $\{\phi(\boldsymbol{x}_v)\}_{v \in \mathcal{V}}$ and $\{\psi(\boldsymbol{x}_v)\}_{v \in \mathcal{V}}$, forming a bipartite graph $\mathcal{S}$ (middle), where the similarity between nodes is defined as $S_{uv} = \text{sim}(u, v) = \phi(\boldsymbol{x}_u)^\top \psi(\boldsymbol{x}_v)$. Due to the clustering objective, nodes that should belong to the same cluster are positioned closer together in the learned graph. These clusters are then identified using spectral biclustering through wKSVD (right).

graphs, as shown in Table 1. HeNCler addresses this by using weighted kernel singular value decomposition (wKSVD) to induce a learned asymmetric similarity graph for both directed and undirected graphs. The dual problem of wKSVD aligns with asymmetric kernel spectral clustering, enabling the interpretation of similarities without homophily. By solving the primal problem directly, HeNCler overcomes computational difficulties and shows superior performance in node clustering tasks within heterophilous graphs.

**Contributions:** Our contributions in this work can be summarized as follows:

- We introduce HeNCler, a kernel spectral biclustering framework designed to *learn* an induced *asymmetric* similarity graph suited for node clustering of heterophilous graphs, applicable to both directed and undirected graphs.

- We develop a primal-dual framework for a generic weighted kernel singular value decomposition (wKSVD) model.

- We show that the dual wKSVD formulation allows for biclustering of bipartite/asymmetric graphs, while we employ a computationally feasible implementation in the primal wKSVD formulation.

- We further generalize our approach with trainable feature mappings, using node and edge decoders, such that the similarity matrix to cluster is learned.

- We train HeNCler in the primal setting and demonstrate its superior performance on the node clustering task for heterophilous attributed graphs. Our implementation is available in supplementary materials.

## 2 PRELIMINARIES AND RELATED WORK

We use lowercase symbols (e.g., $x$) for scalars, lowercase bold (e.g., $\boldsymbol{x}$) for vectors and uppercase bold (e.g., $\boldsymbol{X}$) for matrices. A single entry of a matrix is represented by $X_{ij}$. $\phi(\cdot)$ denotes a mapping and $\boldsymbol{\phi}_v = \phi(\boldsymbol{x}_v)$ represents the mapping of node $v$ in the induced feature space. We represent a graph $\mathcal{G}$ by its vertices (i.e., nodes) $\mathcal{V}$ and edges $\mathcal{E}$, $\mathcal{G} = (\mathcal{V}, \mathcal{E})$, or by its node feature matrix and adjacency matrix $\mathcal{G} = (\boldsymbol{X}, \boldsymbol{A})$. For a bipartite graph, we have $\mathcal{G} = (\mathcal{I}, \mathcal{J}, \mathcal{E})$ or $\mathcal{G} = (\boldsymbol{X}_\mathcal{I}, \boldsymbol{X}_\mathcal{J}, \boldsymbol{S})$ where $S_{ij}$ is the edge weight between nodes $i \in \mathcal{I}$ and $j \in \mathcal{J}$. Note that $\boldsymbol{S}$ is generally asymmetric and rectangular, and that the adjacency matrix of the bipartite graph is given by $\boldsymbol{A} = \begin{bmatrix} \boldsymbol{0} & \boldsymbol{S} \\ \boldsymbol{S}^\top & \boldsymbol{0} \end{bmatrix}$.

**Kernel singular value decomposition** KSVD (Suykens, 2016) sets up a primal-dual framework, based on Lagrange duality, that formulates a variational principle in the primal formulation that corresponds to the matrix singular value decomposition (SVD) in the dual for linear feature maps. By employing non-linear feature mappings or asymmetric kernel functions, this framework allows for non-linear extensions of the SVD problem. The KSVD framework can be applied on data structures such as row and column features, directed graphs, and/or can exploit asymmetric similarity information such as conditional probabilities (He et al., 2023). Interestingly, KSVD often outperforms the similar though symmetric kernel principal component analysis model on tasks where the asymmetry is not immediately apparent (Tao et al., 2024). A different connection is shown in Primal-Attention (Chen et al., 2023), where the authors demonstrate the relation between canonical self-attention, which is asymmetric, and KSVD. They show how to gain computational efficiency by considering a primal equivalent of the attention mechanism.

**Spectral clustering** generalizations have been proposed in many settings. Spectral graph biclustering (Dhillon, 2001) formulates the spectral clustering problem of a bipartite graph $\mathcal{G} = (\mathcal{I}, \mathcal{J}, \boldsymbol{S})$ and shows the equivalence with the SVD of the normalized matrix $\boldsymbol{S}_n = \boldsymbol{D}_1^{-1/2} \boldsymbol{S} \boldsymbol{D}_2^{-1/2}$, where $D_{1,ii} = \sum_j S_{ij}$ and $D_{2,jj} = \sum_i S_{ij}$. Cluster assignments for nodes $\mathcal{I}$ and nodes $\mathcal{J}$ can be inferred from the left and right singular vectors respectively. Further, kernel spectral clustering (KSC) (Alzate & Suykens, 2010) proposes a weighted kernel principal component analysis in which the dual formulation corresponds to the random walks interpretation of the spectral clustering problem. KSC and the aforementioned spectral biclustering formulation lack asymmetry and a primal formulation respectively, which are limitations that our model will address.

**Restricted kernel machines** (RKM) (Suykens, 2017) possess primal and dual model formulations, based on the concept of conjugate feature duality. It is an energy-based framework for (deep) kernel machines, that shows relations with least-squares support vector machines (Suykens et al., 2002) and restricted Boltzmann machines (Salakhutdinov, 2015). The RKM framework encompasses many model classes, including classification, regression, kernel principal component analysis and KSVD, and allows for deep kernel learning (Tonin et al., 2021) and deep kernel learning on graphs (Achten et al., 2024). One possibility to represent the feature maps in RKMs is by means of deep neural networks, e.g., for unsupervised representation learning (Pandey et al., 2021; 2022). RKM models can work in either primal or dual setting, and with decomposition or gradient based algorithms (Achten et al., 2023).

**Homophilous node clustering** methods like MinCutPool (Bianchi et al., 2020) and DMoN (Tsitsulin et al., 2023) introduce unsupervised loss functions within a graph neural network framework. MinCutPool employs a relaxed version of the minimal cut loss applied to the adjacency matrix, while DMoN optimizes the modularity score of clustering labels with respect to the adjacency structure. Both of these methods rely on partitioning the adjacency matrix and inherently assume homophily. Additionally, due to their theoretical underpinnings, these losses are only applicable to undirected graphs. Beyond these adjacency partitioning-based approaches, self-supervised or contrastive methods have also been proposed (You et al., 2020; Hassani & Ahmadi, 2020; Devvrit et al., 2022). These methods typically use graph proximity as their supervision signal, which similarly assumes homophily. For example, S$^3$GC (Devvrit et al., 2022) employs a self-supervised loss based on random walk co-occurrences.

**Heterophilous node clustering** methods typically rely on self-supervised or contrastive techniques. Gong et al. (Gong et al., 2023) propose Sparse Graph Anomaly Detection (SparseGAD), a method that sparsifies graph structures to effectively reduce noise from irrelevant edges and enhance the detection of closely related nodes. This technique reveals underlying node dependencies, accommodating both homophilous and heterophilous relationships. Similarly, HGRL (Chen et al., 2022) employs

self-supervised learning on heterophilous graphs by utilizing graph augmentation techniques to capture global and higher-order structural information. MUSE (Yuan et al., 2023), on the other hand, constructs semantic and contextual views to capture both node-level and neighborhood information for contrastive learning, subsequently integrating these multi-view representations through a fusion controller.

While adjacency partitioning-based methods have demonstrated both theoretical and empirical success for homophilous graphs, they have not been effectively extended to heterophilous graph learning. On the other hand, self-supervised clustering approaches, though promising, often lack a clear clustering interpretation. In the following section, we introduce HeNCler, which bridges these gaps.

## 3 METHOD

**Model motivation** Our approach employs an RKM auto-encoder framework, which has been shown to be effective in unsupervised representation learning by jointly optimizing feature mappings and projection matrices within a kernel-based setting (Pandey et al., 2022). To capture long-range relational dependencies in heterophilous graphs, we utilize a KSVD loss, where a double feature mapping yields a learned asymmetric similarity matrix. To further enhance the cluster-ability of this matrix, we extend the loss function to a weighted KSVD (wKSVD) loss, which not only boosts clustering performance but also offers a spectral graph biclustering interpretation. We next introduce a general wKSVD framework, after which we introduce our HeNCler model that operates in the primal setting while jointly learning the feature mappings end-to-end.

### 3.1 KERNEL SPECTRAL BICLUSTERING WITH ASYMMETRIC SIMILARITIES

Consider a dataset with two, possibly different, input sources $\{\boldsymbol{x}_i\}_{i=1}^n$ and $\{\boldsymbol{z}_j\}_{j=1}^m$, on which we want to define an unsupervised learning task. To this end, we introduce a weighted kernel singular value decomposition model (wKSVD), starting from the following primal optimization problem, which is a weighted variant of the KSVD formulation:

$$\min_{\boldsymbol{U},\boldsymbol{V},\boldsymbol{e},\boldsymbol{r}} J \triangleq \mathrm{Tr}(\boldsymbol{U}^\top \boldsymbol{V}) - \frac{1}{2}\sum_{i=1}^n w_{1,i}\boldsymbol{e}_i^\top \boldsymbol{\Sigma}^{-1}\boldsymbol{e}_i - \frac{1}{2}\sum_{j=1}^m w_{2,j}\boldsymbol{r}_j^\top \boldsymbol{\Sigma}^{-1}\boldsymbol{r}_j$$

$$\text{s.t. } \{\boldsymbol{e}_i = \boldsymbol{U}^\top \phi(\boldsymbol{x}_i),\ \forall i=1,\ldots,n;\quad \boldsymbol{r}_j = \boldsymbol{V}^\top \psi(\boldsymbol{z}_j), \forall j=1,\ldots,m\},\tag{1}$$

with projection matrices $\boldsymbol{U},\boldsymbol{V}\in\mathbb{R}^{d_f\times s}$; strictly positive weighting scalars $w_{1,i}, w_{2,j}$; latent variables $\boldsymbol{e}_i,\boldsymbol{r}_j\in\mathbb{R}^s$; diagonal and positive definite hyperparameter matrix $\boldsymbol{\Sigma}\in\mathbb{R}^{s\times s}$; and centered feature maps $\phi(\cdot):\mathbb{R}^{d_x}\mapsto\mathbb{R}^{d_f}$ and $\psi(\cdot):\mathbb{R}^{d_z}\mapsto\mathbb{R}^{d_f}$; details on centering of the feature maps are provided in Appendix A. The following derivation shows the equivalence with the spectral biclustering problem.

**Proposition 1.** *The solution to the primal problem (1) can be obtained by solving the singular value decomposition of*

$$\boldsymbol{W}_1^{1/2}\boldsymbol{S}\boldsymbol{W}_2^{1/2} = \boldsymbol{H}_e\boldsymbol{\Sigma}\boldsymbol{H}_r^\top,\tag{2}$$

*where $\boldsymbol{W}_1$ and $\boldsymbol{W}_2$ are diagonal matrices such that $W_{1,ii}=w_{1,i}$ and $W_{2,jj}=w_{2,j}$, $\boldsymbol{S}=\boldsymbol{\Phi}\boldsymbol{\Psi}^\top$ is an asymmetric similarity matrix where $S_{ij}=\phi(\boldsymbol{x}_i)^\top\psi(\boldsymbol{z}_j)$, $\boldsymbol{\Phi}=[\phi(\boldsymbol{x}_1)\ldots\phi(\boldsymbol{x}_n)]^\top$, $\boldsymbol{\Psi}=[\psi(\boldsymbol{z}_1)\ldots\psi(\boldsymbol{z}_m)]^\top$, and where $\boldsymbol{H}_e=[\boldsymbol{h}_{e_1}...\boldsymbol{h}_{e_n}]^\top$, and $\boldsymbol{H}_r=[\boldsymbol{h}_{r_1}...\boldsymbol{h}_{r_m}]^\top$ are the left and right singular vectors respectively; and by applying $\boldsymbol{r}_j=\boldsymbol{\Sigma}\boldsymbol{h}_{r_j}/\sqrt{w_{2,j}}$ and $\boldsymbol{e}_i=\boldsymbol{\Sigma}\boldsymbol{h}_{e_i}/\sqrt{w_{1,i}}$.*

*Proof.* We now introduce dual variables $\boldsymbol{h}_{e_i}$ and $\boldsymbol{h}_{r_j}$ using a case of Fenchel-Young inequality (Rockafellar, 1974):

$$\frac{1}{2}w_{1,i}\,\boldsymbol{e}_i^\top\boldsymbol{\Sigma}^{-1}\boldsymbol{e}_i+\frac{1}{2}\boldsymbol{h}_{e_i}^\top\boldsymbol{\Sigma}\boldsymbol{h}_{e_i}\geq\sqrt{w_{1,i}}\,\boldsymbol{e}_i^\top\boldsymbol{h}_{e_i},\quad \frac{1}{2}w_{2,j}\,\boldsymbol{r}_j^\top\boldsymbol{\Sigma}^{-1}\boldsymbol{r}_j+\frac{1}{2}\boldsymbol{h}_{r_j}^\top\boldsymbol{\Sigma}\boldsymbol{h}_{r_j}\geq\sqrt{w_{2,j}}\,\boldsymbol{r}_j^\top\boldsymbol{h}_{r_j},\tag{3}$$

$\forall\boldsymbol{e}_i,\boldsymbol{r}_j,\boldsymbol{h}_{e_i},\boldsymbol{h}_{r_j}\in\mathbb{R}^s$, $\forall w_{1,i},w_{2,j}\in\mathbb{R}_{>0}$, $\forall\boldsymbol{\Sigma}\in\mathbb{R}_{>0}^{s\times s}$. The above inequalities can be verified by writing it in quadratic form: $\frac{1}{2}\begin{bmatrix}\boldsymbol{e}_i^\top & \boldsymbol{h}_{e_i}^\top\end{bmatrix}\begin{bmatrix}w_{1,i}\boldsymbol{\Sigma}^{-1} & -\sqrt{w_{1,i}}\,\boldsymbol{I}_s \\ -\sqrt{w_{1,i}}\,\boldsymbol{I}_s & \boldsymbol{\Sigma}\end{bmatrix}\begin{bmatrix}\boldsymbol{e}_i \\ \boldsymbol{h}_{e_i}\end{bmatrix}\geq 0,\quad\forall i$, with $\boldsymbol{I}_s$ the $s$-dimensional identity matrix, which follows immediately from the Schur complement form:

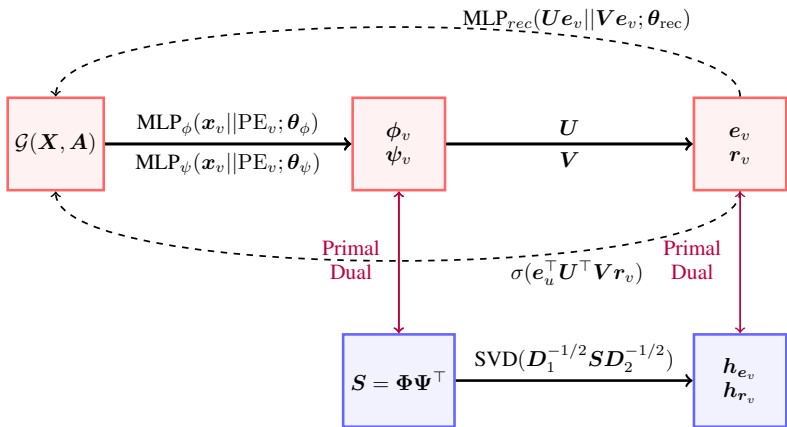

Figure 2: **The HeNCler model**. HeNCler operates in the primal setting (top of the figure in red) and uses a double multilayer perceptron (MLP) to map node representations to a feature space. The obtained representations $\phi_v$ and $\psi_v$ are then projected to latent representations $e_v$ and $r_v$ respectively. The wKSVD loss ensures that these latent representations correspond to the dual equivalent (bottom of the figure in blue) i.e., a biclustering of the asymmetric similarity graph defined by $S$. The node and edge reconstructions (dashed arrows) aid in the feature map learning.

for a matrix $Q = \begin{bmatrix} Q_1 & Q_2 \\ Q_2^\top & Q_3 \end{bmatrix}$, one has $Q \succeq 0$ if and only if $Q_1 \succ 0$ and the Schur complement $Q_3 - Q_2^\top Q_1^{-1} Q_2 \succeq 0$ (Boyd & Vandenberghe, 2004).

By substituting the constraints of (1) and inequalities (3) into the objective function of (1), we obtain an objective in primal and dual variables as an upper bound on the primal objective $\bar{J} \geq J$:

$$\min_{U, V, h_e, h_r} \bar{J} \triangleq \mathrm{Tr}(U^\top V) - \sum_{i=1}^{n} \sqrt{w_{1,i}}\, \phi(x_i)^\top U h_{e_i} + \frac{1}{2} \sum_{i=1}^{n} h_{e_i}^\top \Sigma h_{e_i}$$

$$- \sum_{j=1}^{m} \sqrt{w_{2,j}}\, \psi(z_j)^\top V h_{r_i} + \frac{1}{2} \sum_{j=1}^{m} h_{r_j}^\top \Sigma h_{r_j}. \quad (4)$$

Next, we formulate the stationarity conditions of problem (4):

$$\frac{\partial \bar{J}}{\partial V} = 0 \Rightarrow U = \sum_{j=1}^{m} \sqrt{w_{2,j}}\, \psi(z_j) h_{r_j}^\top, \qquad \frac{\partial \bar{J}}{\partial h_{e_i}} = 0 \Rightarrow \Sigma h_{e_i} = \sqrt{w_{1,i}}\, U^\top \phi(x_i),$$

$$\frac{\partial \bar{J}}{\partial U} = 0 \Rightarrow V = \sum_{i=1}^{n} \sqrt{w_{1,i}}\, \phi(x_i) h_{e_i}^\top, \qquad \frac{\partial \bar{J}}{\partial h_{r_j}} = 0 \Rightarrow \Sigma h_{r_j} = \sqrt{w_{2,j}}\, V^\top \psi(z_j),$$

$$(5)$$

from which we then eliminate the primal variables $U$ and $V$. This yields the eigenvalue problem:

$$\begin{bmatrix} 0 & W_1^{1/2} S W_2^{1/2} \\ W_2^{1/2} S^\top W_1^{1/2} & 0 \end{bmatrix} \begin{bmatrix} H_e \\ H_r \end{bmatrix} = \begin{bmatrix} H_e \\ H_r \end{bmatrix} \Sigma, \quad (6)$$

where $0$ is an all-zeros matrix. Note that, by Lanczos' Theorem (Lanczos, 1958), the above eigenvalue problem is equivalent with (2), and that the stationarity conditions (5) provide the relationships between primal and dual variables, which concludes the proof. □

We have thus shown the connection between the primal (1) and dual formulation (2). Similarly to the KSVD framework, the wKSVD framework can be used for learning with asymmetric kernel functions and/or rectangular data sources. The spectral biclustering problem can now easily be obtained by choosing the weights $w_{1,i}$ and $w_{2,j}$ appropriately.

**Corollary 2.** *Given Proposition 1, and by choosing $\boldsymbol{W}_1$ and $\boldsymbol{W}_2$ to equal $\boldsymbol{D}_1^{-1/2}$ and $\boldsymbol{D}_2^{-1/2}$, where $D_{1,ii} = \sum_j S_{ij}$ and $D_{2,jj} = \sum_i S_{ij}$, we obtain the random walk interpretation $\boldsymbol{D}_1^{-1/2} \boldsymbol{S} \boldsymbol{D}_2^{-1/2} = \boldsymbol{H}_e \boldsymbol{\Sigma} \boldsymbol{H}_r^\top$ of the spectral graph bipartitioning problem for the bipartite graph $\mathcal{S} = (\boldsymbol{\Phi}, \boldsymbol{\Psi}, \boldsymbol{S})$.*

Moreover, the wKSVD framework is more general as, on the one hand, one can use a given similarity matrix (e.g. adjacency matrix of a graph) or (asymmetric) kernel function in the dual, or, on the other hand, one can choose to use explicitly defined (deep) feature maps in both primal or dual.

## 3.2 THE HENCLER MODEL

HeNCler employs the wKSVD framework in a graph setting, where the dataset is a node set $\mathcal{V}$ and where the asymmetry arises from employing to different mappings that operate on the nodes given the entire graph $\mathcal{G} = (\mathcal{X}, \mathcal{A})$. Our method is visualized in Figure 2, where red indicates the primal setting of the framework and blue the dual.

In the preceding subsection, we showed that problem (1) has an equivalent dual problem corresponding to the graph bipartitioning problem, when $w_{1,i}$ and $w_{2,j}$ are chosen to equal the square root of the inverse of the out-degree and in-degree of a similarity graph $\mathcal{S}$ respectively. This similarity graph $\mathcal{S}$ depends on the feature mappings $\phi(\cdot)$ and $\psi(\cdot)$, which for our method does not only depend on the node of interest, but also on the rest of the input graph and the learnable parameters. The mappings for node $v$ thus become $\phi(\boldsymbol{x}_v, \mathcal{G}; \boldsymbol{\theta}_\phi)$ and $\psi(\boldsymbol{x}_v, \mathcal{G}; \boldsymbol{\theta}_\psi)$ and we will ease these notations to $\phi(\boldsymbol{x}_v)$ and $\psi(\boldsymbol{x}_v)$. The ability of our method to learn these feature mappings is an important aspect of our contribution, as a key motivation behind our model is that we need to learn new similarities for clustering heterophilous graphs. The loss function is comprised of three terms: the wKSVD-loss, a node-reconstruction loss, and an edge-reconstruction loss:

$$\mathcal{L}_{\text{wKSVD}}(\boldsymbol{U}, \boldsymbol{V}, \boldsymbol{\Sigma}, \boldsymbol{\theta}_\phi, \boldsymbol{\theta}_\psi) + \mathcal{L}_{\text{NodeRec}}(\boldsymbol{U}, \boldsymbol{V}, \boldsymbol{\theta}_\phi, \boldsymbol{\theta}_\psi, \boldsymbol{\theta}_{\text{rec}}) + \mathcal{L}_{\text{EdgeRec}}(\boldsymbol{U}, \boldsymbol{V}, \boldsymbol{\theta}_\phi, \boldsymbol{\theta}_\psi),$$

where the trainable parameters of the model are in the the multilayer perceptron (MLP) feature maps ($\boldsymbol{\theta}_\phi$ and $\boldsymbol{\theta}_\psi$), the MLP node decoder ($\boldsymbol{\theta}_{\text{rec}}$), in the $\boldsymbol{U}$ and $\boldsymbol{V}$ projection matrices, and in the singular values $\boldsymbol{\Sigma}$. All these parameters are trained end-to-end and we next explain the losses in more detail.

**wKSVD-loss** Instead of solving the SVD in the dual formulation, HeNCler leverages the primal formulation (1) of the wKSVD framework for greater computational efficiency. While equation (1) assumes that the feature maps $\phi(\cdot)$ and $\psi(\cdot)$ are fixed, HeNCler utilizes parametric functions $\phi(\cdot; \boldsymbol{\theta}_\phi)$ and $\psi(\cdot; \boldsymbol{\theta}_\psi)$, enabling it to learn new similarities between nodes. By incorporating regularization terms for these functions and defining the weighting scalars as $w_{1,v} = D_{1,vv}^{-1} = 1/\sum_u \phi(\boldsymbol{x}_v)^\top \psi(\boldsymbol{x}_u)$ and $w_{2,v} = D_{2,vv}^{-1} = 1/\sum_u \phi(\boldsymbol{x}_u)^\top \psi(\boldsymbol{x}_v)$, we derive the wKSVD-loss:

$$\mathcal{L}_{\text{wKSVD}} \triangleq -\sum_{v=1}^{|\mathcal{V}|} D_{1,vv}^{-1} \phi(\boldsymbol{x}_v)^\top \boldsymbol{U} \boldsymbol{\Sigma}^{-1} \boldsymbol{U}^\top \phi(\boldsymbol{x}_v) - \sum_{v=1}^{|\mathcal{V}|} D_{2,vv}^{-1} \psi(\boldsymbol{x}_v)^\top \boldsymbol{V} \boldsymbol{\Sigma}^{-1} \boldsymbol{V}^\top \psi(\boldsymbol{x}_v)$$

$$+ \text{Tr}(\text{U}^\top \text{V}) + \sum_{\text{v}=1}^{|\mathcal{V}|} \sqrt{\text{D}_{1,\text{vv}}^{-1} \text{D}_{2,\text{vv}}^{-1}} \, \phi(\text{x}_\text{v})^\top \psi(\text{x}_\text{v}). \quad (7)$$

The primal formulation of HeNCler (7) can be understood as follows: The first two terms aim to maximize the weighted variance of the learned node representations $\boldsymbol{e}$ and $\boldsymbol{r}$. The third and fourth terms act as regularizers, encouraging asymmetry by penalizing the similarity between $\boldsymbol{U}$ and $\boldsymbol{V}$, and between $\phi(\boldsymbol{x}_v)$ and $\psi(\boldsymbol{x}_v)$, respectively.

For the two feature maps $\phi(\cdot)$ and $\psi(\cdot)$, we employ two MLPs: $\phi(\boldsymbol{x}_v, \mathcal{G}; \boldsymbol{\theta}_\phi) \equiv \text{MLP}_\phi(\boldsymbol{x}_v || \text{PE}_v; \boldsymbol{\theta}_\phi)$ and $\psi(\boldsymbol{x}_v, \mathcal{G}; \boldsymbol{\theta}_\psi) \equiv \text{MLP}_\psi(\boldsymbol{x}_v || \text{PE}_v; \boldsymbol{\theta}_\psi)$. We construct a random walks positional encoding (PE) (Dwivedi et al., 2022) to embed the network's structure and concatenate this encoding with the node attributes. The MLPs have two linear layers with a LeakyReLU activation function in between, followed by a batch normalization layer. The singular values in $\boldsymbol{\Sigma}$ are jointly learned, constrained to lie between 0 and 1, with the additional condition that $\text{Tr}(\boldsymbol{\Sigma}^{-\frac{1}{2}}) = 1$.

**Reconstruction losses** Since the feature maps $\phi(\cdot)$ and $\psi(\cdot)$ need to be learned, an additional loss function beyond the above regularization term is required to effectively optimize the parameters of the MLPs. As the node clustering setting is completely unsupervised, we add a decoder network and

a reconstruction loss. This technique has been proven to be effective for unsupervised learning in the RKM-framework (Pandey et al., 2022), as well as for unsupervised node representation learning (Sun et al., 2021). For heterophilous graphs, we argue that it is particularly important to also reconstruct node features and not only the graph structure.

For the node reconstruction, we first project the $e$ and $r$ variables back to feature space, concatenate these and then map to input space with another MLP. This MLP has also two layers and a leaky ReLU activation function. The hidden layer size is set to the average of the latent dimension and input dimension. With the mean-squared-error as the associated loss, this gives:

$$\mathcal{L}_{\text{NodeRec}} = \frac{1}{|\mathcal{V}|} \sum_{v \in \mathcal{V}} ||\text{MLP}_{\text{rec}}(\boldsymbol{U}\boldsymbol{e}_v||\boldsymbol{V}\boldsymbol{r}_v; \boldsymbol{\theta}_{\text{rec}}) - \boldsymbol{x}_v||^2. \tag{8}$$

To reconstruct edges, we use a simple dot-product decoder $\sigma(\boldsymbol{e}_u^\top \boldsymbol{U}^\top \boldsymbol{V}\boldsymbol{r}_v)$ where $\sigma$ is the sigmoid function. By using the $e$ representation for source nodes and $r$ for target nodes, this reconstruction is asymmetric and can reconstruct directed graphs. We use a binary cross-entropy loss:

$$\mathcal{L}_{\text{EdgeRec}} = \frac{1}{|\mathcal{U}|} \sum_{(u,v) \in \mathcal{U}} \text{BCE}(\sigma(\boldsymbol{e}_u^\top \boldsymbol{U}^\top \boldsymbol{V}\boldsymbol{r}_v), \mathcal{E}_{uv}), \tag{9}$$

where $\mathcal{U}$ is a node-tuple set, resampled every epoch, containing $2|\mathcal{V}|$ positive edges from $\mathcal{E}$ and $2|\mathcal{V}|$ negative edges from $\mathcal{E}^C$, and $\mathcal{E}_{uv} \in \{0, 1\}$ indicates whether an edge $(u, v)$ exist: $(u, v) \in \mathcal{E}$.

**Optimizer, constraints, and cluster assignment** We use Adam (Kingma & Ba, 2015) for the training of all parameters. The batch normalization in the MLP's keeps the wKSVD-loss bounded and the constraints on the singular values is enforced with a softmax function. Cluster assignments are obtained by KMeans clustering on the concatenation of learned $e$ and $r$ node representations.

HeNCler jointly learns the wKSVD projection matrices, $\boldsymbol{U}$ and $\boldsymbol{V}$, along with the feature map parameters, $\boldsymbol{\theta}_\phi$ and $\boldsymbol{\theta}_\psi$. The wKSVD loss improves the cluster-ability of the learned similarity graph, ensuring that $e$ and $r$ function as spectral biclustering embeddings. The two distinct feature maps enable asymmetric learning, effectively capturing potential asymmetric relationships in the data, while the reconstruction losses ensure robust and meaningful representation learning.

Table 2: Dataset statistics of the employed heterophilous graphs.

| Dataset | short | # Nodes | # Edges | # Classes | Directed | $\mathcal{H}(\mathcal{G})$ |
|---|---|---|---|---|---|---|
| Texas | tex | 183 | 325 | 5 | ✓ | 0.000 |
| Cornell | corn | 183 | 298 | 5 | ✓ | 0.150 |
| Wisconsin | wis | 251 | 515 | 5 | ✓ | 0.084 |
| Chameleon | cha | 2,277 | 31,371 | 5 | ✗ | 0.042 |
| Squirrel | squi | 5,201 | 198,353 | 5 | ✗ | 0.031 |
| Roman-empire | rom | 22,662 | 32,927 | 18 | ✗ | 0.021 |
| Minesweeper | mine | 10,000 | 39,402 | 2 | ✗ | 0.009 |
| Tolokers | tol | 11,758 | 519,000 | 2 | ✗ | 0.180 |

# 4 EXPERIMENTS

**Datasets** We assess the performance of HeNCler on heterophilous attributed graphs that are available in literature. we use Texas, Cornell, and Wisconsin (Pei et al., 2020)[1], which are directed webpage networks where edges encode hyperlinks between pages. Next, we use Chameleon and Squirrel (Rozemberczki et al., 2021), which are undirected Wikipedia webpage networks where edges encode mutual links. We further assess our model on the undirected graphs: Roman-empire, Minesweeper, and Tolokers (Platonov et al., 2023), which are a graph representation of a Wikipedia article, a grid graph based on the minesweeper game, and a crowd-sourcing network respectively. We include

---

[1]http://www.cs.cmu.edu/afs/cs.cmu.edu/project/theo-11/www/wwkb

Table 3: Experimental results on heterophilous graphs. We report NMI and F1 scores for 10 runs (mean $\pm$ standard deviation), where higher values indicate better performance. The best results for each metric are highlighted in bold.

| | | | Baselines | | | | Ours |
|---|---|---|---|---|---|---|---|
| Dataset | | KMeans | MinCutPool | DMoN | S$^3$GC | MUSE | HeNCler |
| tex | NMI | $4.97_{\pm1.00}$ | $11.60_{\pm2.19}$ | $9.06_{\pm2.11}$ | $11.56_{\pm1.46}$ | $39.23_{\pm4.91}$ | $\mathbf{43.65}_{\pm2.52}$ |
| | F1 | $59.27_{\pm0.83}$ | $55.26_{\pm0.56}$ | $47.76_{\pm4.79}$ | $43.69_{\pm2.74}$ | $65.96_{\pm3.52}$ | $\mathbf{71.39}_{\pm2.16}$ |
| corn | NMI | $5.42_{\pm2.04}$ | $17.04_{\pm1.61}$ | $12.49_{\pm2.51}$ | $14.48_{\pm1.79}$ | $38.99_{\pm2.73}$ | $\mathbf{41.52}_{\pm4.35}$ |
| | F1 | $52.97_{\pm0.24}$ | $51.21_{\pm5.06}$ | $43.83_{\pm6.23}$ | $33.13_{\pm0.83}$ | $60.58_{\pm3.61}$ | $\mathbf{63.40}_{\pm3.67}$ |
| wis | NMI | $6.84_{\pm4.39}$ | $13.38_{\pm2.36}$ | $12.56_{\pm1.23}$ | $13.07_{\pm0.61}$ | $39.71_{\pm2.22}$ | $\mathbf{47.13}_{\pm1.76}$ |
| | F1 | $56.16_{\pm0.58}$ | $55.63_{\pm2.96}$ | $45.72_{\pm7.85}$ | $31.71_{\pm2.25}$ | $58.94_{\pm3.09}$ | $\mathbf{68.30}_{\pm2.17}$ |
| cha | NMI | $0.44_{\pm0.11}$ | $11.88_{\pm1.99}$ | $12.87_{\pm1.86}$ | $15.83_{\pm0.26}$ | $23.06_{\pm0.28}$ | $\mathbf{23.89}_{\pm0.84}$ |
| | F1 | $\mathbf{53.23}_{\pm0.07}$ | $50.40_{\pm5.65}$ | $45.05_{\pm4.30}$ | $36.51_{\pm0.24}$ | $52.10_{\pm0.48}$ | $44.14_{\pm1.83}$ |
| squi | NMI | $1.40_{\pm2.12}$ | $6.35_{\pm0.32}$ | $3.08_{\pm0.38}$ | $3.83_{\pm0.11}$ | $8.30_{\pm0.23}$ | $\mathbf{9.67}_{\pm0.13}$ |
| | F1 | $54.05_{\pm2.72}$ | $\mathbf{55.26}_{\pm0.57}$ | $49.21_{\pm2.74}$ | $35.08_{\pm0.18}$ | $50.07_{\pm5.99}$ | $36.51_{\pm2.39}$ |
| rom | NMI | $35.20_{\pm1.79}$ | $9.97_{\pm2.02}$ | $13.14_{\pm0.53}$ | $14.48_{\pm0.21}$ | $\mathbf{40.50}_{\pm0.73}$ | $36.99_{\pm0.61}$ |
| | F1 | $37.17_{\pm2.12}$ | $\mathbf{42.19}_{\pm0.26}$ | $22.69_{\pm3.91}$ | $17.76_{\pm0.53}$ | $38.34_{\pm0.35}$ | $35.43_{\pm1.07}$ |
| mine | NMI | $0.02_{\pm0.02}$ | $6.16_{\pm2.17}$ | $\mathbf{6.87}_{\pm2.91}$ | $6.53_{\pm0.17}$ | $0.06_{\pm0.01}$ | $0.06_{\pm0.00}$ |
| | F1 | $73.63_{\pm3.58}$ | $71.76_{\pm8.86}$ | $70.42_{\pm9.47}$ | $48.78_{\pm0.63}$ | $75.77_{\pm2.24}$ | $\mathbf{76.48}_{\pm1.56}$ |
| tol | NMI | $3.04_{\pm2.83}$ | $6.68_{\pm0.98}$ | $6.69_{\pm0.20}$ | $5.99_{\pm0.05}$ | $6.67_{\pm0.55}$ | $\mathbf{6.73}_{\pm0.59}$ |
| | F1 | $65.56_{\pm10.49}$ | $72.10_{\pm10.38}$ | $67.87_{\pm4.74}$ | $59.17_{\pm0.27}$ | $73.56_{\pm1.94}$ | $\mathbf{73.66}_{\pm2.10}$ |

experimental results for additional homophilous datasets in Appendix B. The dataset statistics can be consulted in Table 2, where the class insensitive edge homophily ratio $\mathcal{H}(\mathcal{G})$ (Lim et al., 2021) is a homophily measure.

**Model selection and metrics** Model selection in this unsupervised setting is non-trivial, and the best metric depends on the task at hand. Therefore, this is not the scope of this paper and we assess our model agnostically to the model selection, and fairly w.r.t. to the baselines. We fix the hyperparameter configuration of the models across all datasets. We train for a fixed number of epochs and keep track of the evaluation metrics to report the best observed result. We repeat the training process 10 times and report average best results with standard deviations. We report the normalized mutual information (NMI) and pairwise F1-scores, based on the class labels.

**Baselines and hyperparameters** We compare our model against several methods, including a simple KMeans based on node attributes, adjacency partitioning-based approaches such as MinCutPool (Bianchi et al., 2020) and DMoN (Tsitsulin et al., 2023), as well as S$^3$GC (Devvrit et al., 2022) and MUSE (Yuan et al., 2023), which represent the current state-of-the-art in homophilous and heterophilous node clustering, respectively. For HeNCler, we fix the hyperparameters to: MLP hidden dimensions 256, output dimensions 128, latent dimension $s = 2 \times$ #classes, learning rate 0.01, and epochs 300. For the baselines, we used their code implementations and the default hyperparameter settings as proposed by the authors. The number of clusters to infer is set to the number of classes cfr. Table 2 for all methods. The experiments are run on a Nvidia V100 GPU.

**Experimental results** Table 3 presents the experimental results for heterophilous graphs. HeNCler consistently demonstrates superior performance, significantly outperforming KMeans, MinCutPool, DMoN, S$^3$GC, and MUSE, especially on the directed graphs. For undirected graphs, HeNCler also shows strong results, achieving the best performance in 5 out of 10 cases, compared to KMeans (1/10), MinCutPool (2/10), DMoN (1/10), S$^3$GC (0/10), and MUSE (1/10). These results highlight HeNCler's versatility and effectiveness in handling heterophilous graph structures.

**Ablation studies** We conduct several ablation studies, presented in Table 4. The 'Undirected' variant refers to a simplified, symmetric version of the model that uses a single MLP for both the $\phi(\cdot)$ and $\psi(\cdot)$ mappings, i.e., $\phi(\cdot) \equiv \psi(\cdot)$. In this version, the model loses its asymmetry. The

Table 4: Ablation study results. We report mean NMI and F1 scores for 10 runs (higher is better) for different model configurations. Best results are highlighted in bold.

| Metric | tex | | corn | | cha | | rom | | tol | |
|---|---|---|---|---|---|---|---|---|---|---|
| | NMI | F1 | NMI | F1 | NMI | F1 | NMI | F1 | NMI | F1 |
| Undirected | 27.58 | 65.20 | 18.12 | 53.69 | 19.91 | 44.08 | 33.17 | 33.57 | 6.33 | 73.89 |
| Reconstr only | 29.54 | 66.64 | 27.76 | 54.70 | 22.02 | 43.42 | **40.05** | 35.16 | 6.18 | 68.60 |
| wKSVD only | 31.64 | 62.83 | 20.63 | 47.12 | 22.60 | 42.98 | 35.99 | 35.30 | 4.42 | 68.45 |
| HeNCler | **43.65** | **71.39** | **41.52** | **63.40** | **23.89** | **44.14** | 36.99 | **35.43** | **6.73** | 73.66 |

'wKSVD only' and 'Reconstr only' variations reflect models that incorporate only the wKSVD loss ($\mathcal{L}_{\text{wKSVD}}$) and the reconstruction losses ($\mathcal{L}_{\text{NodeRec}} + \mathcal{L}_{\text{EdgeRec}}$), respectively. Interestingly, as shown in Table 4, even for undirected graphs, introducing asymmetry in HeNCler enhances clustering performance. Furthermore, all loss components are shown to contribute positively to HeNCler's overall performance. For a comprehensive analysis, including results across all datasets and standard deviations, we refer the reader to Table 7 in Appendix B.

## 5 DISCUSSION

A key motivation behind HeNCler is to learn a new graph representation where nodes belonging to the same cluster are positioned closer together, driven by the clustering objective. This results in spectral biclustering embeddings that exhibit improved cluster-ability. Note that HeNCler uses KMeans to obtain cluster assignments. Therefore, the comparisons between HeNCler and KMeans, as shown in Tables 3 and 6 , demonstrate that our model enhances the cluster-ability of the node representations relative to the original input features.

The asymmetry in HeNCler eliminates the undirected constraints of traditional adjacency partitioning-based models, enabling superior performance on directed graphs, as shown in Table 3. Furthermore, our ablation study in Table 4 shows that, while most of the performance on undirected graphs stem from the graph learning component, HeNCler is additionally able to capture and learn meaningful asymmetric information. This capacity to extract valuable asymmetric insights from symmetric data is a common occurrence in KSVD frameworks (He et al., 2023; Tao et al., 2024). Importantly, thanks to the added performance boost from asymmetry, on top of the benefits from similarity learning, HeNCler outperforms state-of-the-art models, even when applied to undirected graphs.

We visualize the learned similarity matrix $S = \Phi\Psi^\top$ for two datasets in Figure 3. These matrices are generally asymmetric, with the asymmetry particularly pronounced in the directed graph of the Wisconsin dataset. In contrast, the Roman-Empire dataset, which is represented by an undirected graph, exhibits less asymmetry in the learned similarity matrix. This demonstrates the adaptability of HeNCler to handle both directed and undirected graphs. Further, given the observable block structures, the learned similarities are meaningful w.r.t. to the ground truth node labels. Note however that our model operates in the primal setting and directly projects

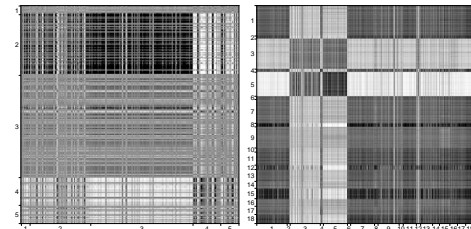

Figure 3: The learned matrix $S = \Phi\Psi^\top$ for the Wisconsin (left) and Roman-empire (right) dataset. Rows and columns are grouped according to ground-truth node labels.

the learned mappings $\phi$ and $\psi$ to their final embeddings $e$ and $r$ using $U$ and $V$ respectively, avoiding quadratic space complexity and cubic time complexity of the SVD. This is the motivation of employing a kernel based method, and exploiting the primal-dual framework that comes with it. In fact, the matrices in Figure 3 are only constructed for the sake of this visualization.

**Computational complexity** The space and time complexity of the current implementation of HeNCler are both linear w.r.t. the number of nodes $\mathcal{O}(|\mathcal{V}|)$. Whereas MinCutPool and DMoN need

all the node attributes in memory to calculate the loss w.r.t. the full adjacency matrix, HeNCler is easily adaptable to work with minibatches which reduces space complexity to the minibatch size $\mathcal{O}(|\mathcal{B}|)$. Although HeNCler relies on edge reconstruction, the edge sampling avoids quadratic complexity w.r.t. number of nodes, and is specifically designed to scale with the number of nodes, rather than the number of edges. Assuming the graphs are sparse, we add an overview of space and time complexity w.r.t. the number of nodes and edges for all methods in Table 1. A detailed table with measured computation times is provided in Appendix C.

## 6 CONCLUSION AND FUTURE WORK

We tackle three limitations of current node clustering algorithms, that prevent these methods from effectively clustering nodes in heterophilous graphs: they assume homophily in their loss, they are only defined for undirected graphs and/or they lack a specific focus on clustering.

To this end, we introduce a weighted kernel SVD framework and harness its primal-dual equivalences. HeNCler relies on the dual interpretation for its theoretical motivation, while it benefits from the computational advantages of its implementation in the primal. In an end-to-end fashion, it learns new similarities, which are asymmetric where necessary, and node embeddings resulting from the spectral biclustering interpretation of these learned similarities. As empirical evidence shows, our approach effectively eliminates the aforementioned limitations, significantly outperforming current state-of-the-art alternatives.

HeNCler is the first heterophilous node clustering model that does not rely on contrastive learning techniques. Future research could explore the integration of contrastive learning into HeNCler, potentially combining the strengths of both approaches. Another next step can be to investigate how to do the cluster assignments in a graph pooling setting (i.e., differentiable graph coarsening), to enable end-to-end learning for downstream graph prediction tasks.

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

## A    NOTE ON FEATURE MAP CENTERING

In the wKSVD framework, we assume that the feature maps are centered. More precisely, given two arbitrary mappings $\phi(\cdot)$ and $\psi(\cdot)$, the centered mappings are obtained by subtracting the weighted mean:

$$\phi_c(\boldsymbol{x}_i) = \phi(\boldsymbol{x}_i) - \frac{\sum_{k=1}^{n} w_{1,k}\ \phi(\boldsymbol{x}_k)}{\sum_{k=1}^{n} w_{1,k}},$$

$$\psi_c(\boldsymbol{z}_j) = \psi(\boldsymbol{z}_j) - \frac{\sum_{l=1}^{m} w_{2,l}\ \psi(\boldsymbol{z}_l)}{\sum_{l=1}^{m} w_{2,l}}.$$

Although we use the primal formulation in this paper, we next show how to obtain this centering in the dual for the sake of completeness. When using a kernel function or a given similarity matrix, one has no access to the explicit mappings and has to do an equivalently centering in the dual using:

$$\boldsymbol{S}_c = \boldsymbol{M}_1 \boldsymbol{S} \boldsymbol{M}_2^{\top},$$

where $\boldsymbol{M}_1$ and $\boldsymbol{M}_2$ are the centering matrices:

$$\boldsymbol{M}_1 = \boldsymbol{I}_n - \frac{1}{\mathbf{1}_n^{\top} \boldsymbol{W}_1 \mathbf{1}_n} \mathbf{1}_n \mathbf{1}_n^{\top} \boldsymbol{W}_1$$

$$\boldsymbol{M}_2 = \boldsymbol{I}_m - \frac{1}{\mathbf{1}_m^{\top} \boldsymbol{W}_2 \mathbf{1}_m} \mathbf{1}_m \mathbf{1}_m^{\top} \boldsymbol{W}_2,$$

with $\boldsymbol{I}_n$ and $\mathbf{1}_n$ a $n \times n$ identity matrix and a $n$-dimensional all-ones vector respectively. We omit the subscript $c$ in the paper and assume the feature maps are always centered. Note that this can easily be achieved in the implementations by using the above equations.

## B    ADDITIONAL EXPERIMENTS

**Homophilous experiments**   Although our work primarily focuses on heterophilous graphs, we further evaluate our model on homophilous citation networks Cora, Citeseer, and PubMed (Sen et al., 2008; Yang et al., 2016). The dataset statistics can be consulted in Table 5. We employ the same experimental setup as for the heterophilous datasets and report the experimental results in Table 6. While S$^3$GC achieves the best overall performance due to its alignment with the homophily assumption, HeNCler outperforms adjacency partitioning methods like MinCutPool and DMoN. Additionally, HeNCler demonstrates competitive performance with MUSE, the state-of-the-art in heterophilous node clustering, further validating its robustness across different graph types.

Table 5: Dataset statistics of the employed homophilous graphs.

| Dataset | short | # Nodes | # Edges | # Classes | Directed | $\mathcal{H}(\mathcal{G})$ |
|---------|-------|---------|---------|-----------|----------|------|
| Cora | cora | 2,708 | 5,278 | 7 | ✗ | 0.765 |
| CiteSeer | cite | 3,327 | 4,614 | 6 | ✗ | 0.627 |
| Pubmed | pub | 19,717 | 44,325 | 3 | ✗ | 0.664 |

**Comprehensive ablation study**   We provide the full ablation study results in Table 7, including all datasets and standard deviations. We compare HeNCler with three simplified versions. 'Undirected' reflects an undirected variant of the model with a single MLP decoder. 'wKSVD only' and 'Reconstr only' is the model where only the wKSVD loss $\mathcal{L}_{wKSVD}$ and the reconstruction losses $\mathcal{L}_{\text{NodeRec}} + \mathcal{L}_{\text{EdgeRec}}$ are used respectively. We observe that HeNCler performs better than its undirected version, even for undirected graphs, and that all loss terms contribute to HeNCler's performance.

## C    COMPUTATION TIMES

We trained MinCutPool, DMoN, and HeNCler for 300 iterations; and S$^3$GC for 30 iterations on a Nvidia V100 GPU, and report the computation times in Table 8. Figure 4 visualises these result w.r.t. the number of nodes in the graph, showing the linear time complexity of HeNCler and that it is insensitive to the number of edges. We conclude that HeNCler demonstrates fast computation times.

Table 6: Experimental results on homophilous graphs. We report NMI and F1 scores for 10 runs (mean $\pm$ standard deviation), where higher values indicate better performance. The best results for each metric are highlighted in bold.

| Dataset | | Baselines | | | | | Ours |
|---|---|---|---|---|---|---|---|
| | | KMeans | MinCutPool | DMoN | S$^3$GC | MUSE | HeNCler |
| cora | NMI | $35.0_{\pm 3.21}$ | $49.0_{\pm 2.24}$ | $51.7_{\pm 1.63}$ | $\mathbf{53.62}_{\pm 0.55}$ | $36.45_{\pm 2.71}$ | $38.81_{\pm 2.26}$ |
| | F1 | $36.0_{\pm 2.12}$ | $47.1_{\pm 1.78}$ | $51.8_{\pm 2.02}$ | $\mathbf{60.12}_{\pm 0.46}$ | $50.78_{\pm 2.79}$ | $47.93_{\pm 2.60}$ |
| cite | NMI | $19.9_{\pm 2.90}$ | $29.5_{\pm 3.21}$ | $30.3_{\pm 1.09}$ | $\mathbf{43.56}_{\pm 0.65}$ | $39.03_{\pm 1.99}$ | $34.83_{\pm 2.21}$ |
| | F1 | $39.4_{\pm 3.07}$ | $47.1_{\pm 1.21}$ | $57.4_{\pm 3.42}$ | $\mathbf{64.12}_{\pm 0.28}$ | $52.89_{\pm 1.68}$ | $48.70_{\pm 2.79}$ |
| pub | NMI | $31.4_{\pm 2.18}$ | $21.4_{\pm 1.46}$ | $25.7_{\pm 2.46}$ | $31.01_{\pm 2.35}$ | $\mathbf{36.09}_{\pm 3.26}$ | $27.26_{\pm 1.72}$ |
| | F1 | $59.2_{\pm 2.32}$ | $44.5_{\pm 2.47}$ | $34.3_{\pm 2.05}$ | $\mathbf{69.12}_{\pm 1.39}$ | $61.26_{\pm 1.50}$ | $51.17_{\pm 1.75}$ |

Table 7: Full Ablation study results. We report NMI and F1 scores for 10 runs (mean $\pm$ standard deviation in %) where higher is better. Best results are highlighted in bold.

| dataset | | ablations | | | full model |
|---|---|---|---|---|---|
| | | Undirected | Reconstr only | wKSVD only | HeNCler |
| tex | NMI | $27.58_{\pm 4.75}$ | $29.54_{\pm 2.27}$ | $31.64_{\pm 2.14}$ | $\mathbf{43.65}_{\pm 2.52}$ |
| | F1 | $65.20_{\pm 2.06}$ | $66.64_{\pm 1.83}$ | $62.83_{\pm 3.91}$ | $\mathbf{71.39}_{\pm 2.16}$ |
| corn | NMI | $18.12_{\pm 2.57}$ | $27.76_{\pm 3.29}$ | $20.63_{\pm 5.92}$ | $\mathbf{41.52}_{\pm 4.35}$ |
| | F1 | $53.69_{\pm 0.98}$ | $54.70_{\pm 1.88}$ | $47.12_{\pm 2.61}$ | $\mathbf{63.40}_{\pm 3.67}$ |
| wis | NMI | $25.08_{\pm 3.54}$ | $34.65_{\pm 1.86}$ | $39.86_{\pm 4.63}$ | $\mathbf{47.13}_{\pm 1.76}$ |
| | F1 | $57.13_{\pm 1.34}$ | $62.28_{\pm 1.58}$ | $63.60_{\pm 2.46}$ | $\mathbf{68.30}_{\pm 2.17}$ |
| cha | NMI | $19.91_{\pm 0.48}$ | $22.02_{\pm 0.25}$ | $22.60_{\pm 0.57}$ | $\mathbf{23.89}_{\pm 0.84}$ |
| | F1 | $44.08_{\pm 1.79}$ | $43.42_{\pm 1.63}$ | $42.98_{\pm 0.37}$ | $\mathbf{44.14}_{\pm 1.83}$ |
| squi | NMI | $9.59_{\pm 0.21}$ | $9.59_{\pm 0.27}$ | $9.56_{\pm 0.19}$ | $\mathbf{9.67}_{\pm 0.13}$ |
| | F1 | $\mathbf{55.43}_{\pm 0.03}$ | $53.74_{\pm 3.77}$ | $36.42_{\pm 1.85}$ | $36.51_{\pm 2.39}$ |
| rom | NMI | $33.17_{\pm 1.25}$ | $\mathbf{40.05}_{\pm 0.82}$ | $35.99_{\pm 0.95}$ | $36.99_{\pm 0.61}$ |
| | F1 | $33.57_{\pm 2.15}$ | $35.16_{\pm 1.34}$ | $35.30_{\pm 0.97}$ | $\mathbf{35.43}_{\pm 1.07}$ |
| mine | NMI | $\mathbf{0.08}_{\pm 0.02}$ | $0.07_{\pm 0.02}$ | $0.04_{\pm 0.01}$ | $0.06_{\pm 0.00}$ |
| | F1 | $76.15_{\pm 2.25}$ | $76.05_{\pm 2.16}$ | $73.77_{\pm 3.40}$ | $\mathbf{76.48}_{\pm 1.56}$ |
| tol | NMI | $6.33_{\pm 0.94}$ | $6.18_{\pm 0.67}$ | $4.42_{\pm 0.54}$ | $\mathbf{6.73}_{\pm 0.59}$ |
| | F1 | $73.89_{\pm 4.00}$ | $68.60_{\pm 5.95}$ | $68.45_{\pm 7.57}$ | $\mathbf{73.66}_{\pm 2.10}$ |

Table 8: Computation times in seconds.

| DATASET | BASELINES | | | OURS |
|---|---|---|---|---|
| | MINCUTP. | DMON | S$^3$GC | HENCLER |
| CHAMELEON | 8 | 20 | 89 | 24 |
| SQUIRREL | 14 | 86 | 105 | 49 |
| ROMAN-EMPIRE | 67 | 71 | 312 | 57 |
| AMAZON-RATING | 109 | 93 | 195 | 63 |
| MINESWEEPER | 55 | 27 | 98 | 21 |
| TOLOKERS | 71 | 198 | 100 | 35 |
| QUESTIONS | 340 | 215 | 217 | 125 |

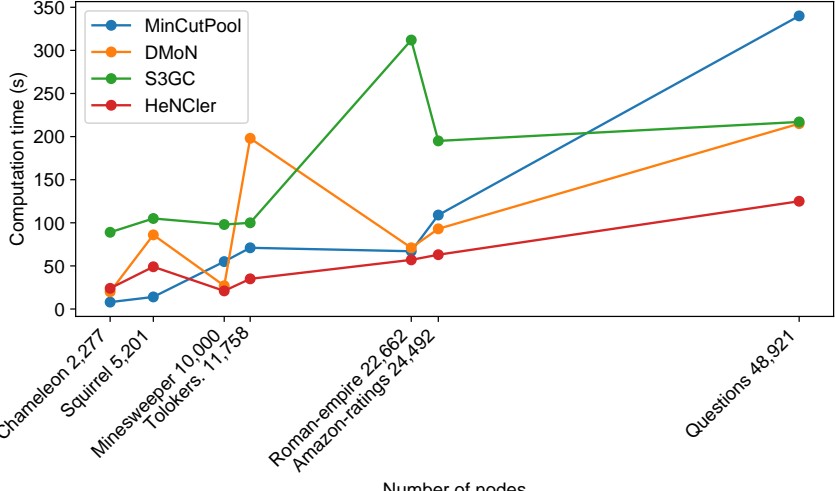

Figure 4: Computation times of MinCutPool, DMoN, S$^3$GC, and HeNCler w.r.t. the number of nodes of the datasets. We observe that HeNCler scales linearly with the number of nodes, and that it is not sensitive to the number of edges, as opposed to DMoN, showing a significant peak for the Tolokers dataset due the large number of edges in this graph.

