# OpenReview forum: "HeNCler: Node Clustering in Heterophilous Graphs via Learned Asymmetric Similarity"
_ICLR.cc/2025/Conference — Submitted to ICLR 2025_

### Official Review · Reviewer_AvNN · 2024-10-27

**Soundness:** 3
**Presentation:** 3
**Contribution:** 2
**Rating:** 1
**Confidence:** 5

**Summary:**

HeNCler employs weighted kernel singular value decomposition (wKSVD) to learn an induced asymmetric similarity graph, avoiding reliance on homophily and enabling more effective clustering in heterophilous settings.

**Strengths:**

1.The motivation is clear.
2. The proposed method is sound to me.
3. The experimental results show the effectiveness of the proposed method.

**Weaknesses:**

1. The technique contribution is limited. The main contribution is the construction of asymmetric similarity, while the other method part is a combination of existing components.
2. The authors ignore many existing works, like [1] and [2].
3. The comparison methods are not enough to show their effectiveness. For example, the newest baselines S3GC and MUSE are from 2022 and 2023 respectively. Besides, S3GC focuses on homophilic and large graphs and MUSE focuses on multi-view graphs.
4. The application is limited. This work seems to focus only on heterophyllous graphs. From Table 6, it achieves poor performance on homophilous graphs. However, the homophily is unknown without the labeled data.

[1] Dong, Y., Dupty, M. H., Deng, L., Liu, Z., Goh, Y. L., & Lee, W. S. (2024). Differentiable Cluster Graph Neural Network. arXiv preprint arXiv:2405.16185.
[2] Jiang, Mengying, et al. "GCN-SL: Graph Convolutional Networks with Structure Learning for Graphs under Heterophily." arXiv preprint arXiv:2105.13795 (2021).

**Questions:**

1. There are a number of works that performs clustering on heterophilic graph, which should be cited or compared. To name a few, [1-4]
2. In fact, some existing methods also learn new similarity matrices from data for heterophilic graph [1-2], which share a similar idea with the proposed method.


Refs:
[1] Beyond Homophily: Reconstructing Structure for Graph-agnostic Clustering, ICML 2023.
[2] Robust Graph Structure Learning under Heterophily, arXiv 2024.
[3] Homophily-enhanced structure learning for graph clustering. Proceedings of the 32nd ACM International Conference on Information and Knowledge Management. 2023.
[4] Homophily-Related: Adaptive Hybrid Graph Filter for Multi-View Graph Clustering. AAAI 2024.

---

> ### Author Response · Authors · 2024-11-25
>
> Our work generalizes a spectral clustering framework to adapt to heterophilous and asymmetrical graphs, with a theoretical foundation rooted in a primal-dual framework. This approach is markedly different from existing work.
>
> As we mentioned in our responses to other reviewers, we will include additional comparisons with more general representation learning methods for heterophilous graphs in our final paper. However, the most relevant comparison remains with MUSE, the current state-of-the-art for purely unsupervised node clustering in heterophilous graphs. Contrary to your statement, MUSE does not specifically focus on multi-view graphs; rather, it constructs two views as part of its approach.

---

> > ### Comment · Reviewer_AvNN · 2024-11-25
> >
> > Thanks for the response.

---

> ### Comment · Area_Chair_UMwD · 2024-11-25
>
> Could please acknowledge and respond to the rebuttal.

---

### Official Review · Reviewer_ZV7T · 2024-11-01

**Soundness:** 2
**Presentation:** 3
**Contribution:** 1
**Rating:** 3
**Confidence:** 4

**Summary:**

The paper introduces HeNCler, a novel approach for clustering nodes in heterophilous graphs. Traditional clustering methods often assume homophily, where nodes with similar labels are more likely to be connected. HeNCler addresses this challenge by learning a similarity graph through a clustering-specific objective based on weighted kernel singular value decomposition. The authors claim that HeNCler significantly improves node clustering performance in heterophilous graph settings, highlighting the advantage of its asymmetric graph-learning framework.

**Strengths:**

1. The paper addresses a significant and under-explored problem in the field of node clustering for heterophilous graphs. This is a practical and relevant issue, as many real-world graphs exhibit heterophily.
2. Leveraging weighted kernel singular value decomposition to learn a similarity graph for deep node clustering appears to be novel.
3. The whole objective combines the proposed wKSVD loss with a graph autoencoder. Ablation studies suggest that both components contribute to the overall performance of HeNCler.

**Weaknesses:**

1. The relationship between the proposed wKSVD loss and the characteristics of heterophilous graphs is unclear, which limits the soundness of the method. To my understanding, the most important part of wKSVD is its asymmetric similarity measure, which applies different projections to the pairs. However, it is a pretty trivial operation and this paper does not provide a in-depth explanation of how it helps in the context of heterophilous graphs.
2. While the paper claims significant performance improvements, it does not compare with strong baselines or state-of-the-art methods for heterophilous graphs. There are various unsupervised heterophilous graph representation learning methods that could serve as competitive baselines, but the only competitor designed for heterophilous graphs is MUSE.
3. I don't agree with the claimed computational complexity improvement of HeNCler. According to _Optimizer, constraints, and cluster assignment_, cluster assignments are obtained by KMeans clustering on the final embeddings, so you need to include the complexity of KMenas in the overall complexity analysis.
4. The wKSVD-loss is based on the primal formulation of wKSVD, so it's unclear how most of Section 3.1 relates to the proposed method.

**Questions:**

1. Please provide more theoretical evidence to strengthen the effectiveness of wKSVD for heterophilous graphs.
2. Consider removing the dual formulation if it's unrelated to the proposed method.
3. Please add additional experiments comparing HeNCler with state-of-the-art methods for heterophilous graph clustering or representation learning.

---

> ### Author Response · Authors · 2024-11-25
>
> **Q1: Theoretical Motivation**
>
> The motivation behind HeNCler is to propose a spectral-based method, as such methods have proven theoretical and empirical success in node clustering tasks. Rather than specifically targeting heterophily, we alleviate the homophily assumption inherent in many spectral-based methods. By learning a new similarity graph, HeNCler can adapt to heterophilous settings.
>
> **Q2: Dual formulation**
>
> The dual formulation is an essential part of our theoretical framework and underpins the model’s motivation. Section 3.1 introduces the new similarity graph and proves the equivalence of the primal formulation to the spectral clustering objective. This is critical to HeNCler’s soundness and demonstrates that the wKSVD-based objective is not arbitrary but deeply tied to spectral clustering principles. We argue that this part of the paper is highly relevant to the soundness of our method and its theoretical understanding.
>
> **Q3: Additional Comparisons**
>
> We compare HeNCler with MUSE, the current state-of-the-art for purely unsupervised node clustering in heterophilous graphs. In response to your feedback, we will include comparisons with additional methods, based on the references provided by the other reviewers. These experiments will help contextualize HeNCler’s performance among broader heterophilous representation learning approaches.

---

> > ### Comment · Reviewer_ZV7T · 2024-11-26
> >
> > I appreciate the author's response and now understand the aim of the dual formulation. However, why the proposed wKSVD loss is effective for heterophilic graphs is still unclear. I've read the responses from other reviewers and share some of their concerns, so I'll maintain my original score.

---

> ### Comment · Area_Chair_UMwD · 2024-11-25
>
> Could please acknowledge and respond to the rebuttal.

---

### Official Review · Reviewer_8wDQ · 2024-11-03

**Soundness:** 3
**Presentation:** 3
**Contribution:** 3
**Rating:** 6
**Confidence:** 3

**Summary:**

Focusing on node clustering on heterophilous graph, this paper introduces an RMK-based framework for learning an asymmetric similarity. By incorporating wKSVD framework into graph clustering with two reconstruction loss, HeCler can optimize a bipartite graph with theoretical interpretability.  The experimental results and empirical analysis can demonstrate the effectiveness.

**Strengths:**

1. Novelty: Although the overall idea of learning a similarity matrix for heterophilous graph learning is not new, this paper proposed an interesting framework by observing the advantage of RMK framework.
2. The theoretical support brings interpretability to this framework.
3. The discussion could empirically interpret how the method works.

**Weaknesses:**

1. One weakness is the lack of literature on heterophyllous graph learning. The authors only list three works for heterophilous node clustering. Other methods for heterophyllous graph representation learning are suggested to be included on top of node clustering (such as heterophilous node classification, multi-view heterophilous node clustering, etc).

2. Analysis about the experimental results are somehow insufficient. The experimental results on undirected graphs seems not as good as directed graphs, such as Chamelon, squirrel. The authors should analyze this. In addition, the NMI in these datasets are very low (such as 9.67, 0.06, 6.73), which phenomenon indicates that the model even does not learn any clustering information. How to explain this?

**Questions:**

1. Can you provide a detailed analysis of why your method performs differently on directed vs undirected graphs, particularly for datasets like Chameleon and Squirrel?
2. The NMI scores for some datasets (e.g. 9.67, 0.06, 6.73) are very low. What might these low scores indicate about the model's performance or limitations on these particular datasets? Are there any insights to be gained from comparing these results to other methods or baseline performance on these challenging datasets?

---

> ### Author Response · Authors · 2024-11-25
>
> Thank you for pointing out the need for broader literature coverage.
> We will revise the Related Work section to include additional references on heterophilous graph representation learning, such as heterophilous node classification, to provide a more comprehensive context for our contributions. We will better point out the unique challenges of purely unsupervised heterophilous node clustering and the limitations of these generally supervised methods.
>
> **Q1: Performance on Directed vs. Undirected Graphs**
>
> The core motivation behind our asymmetrical model is to handle both directed and undirected datasets, as opposed to existing spectral-based methods that rely heavily on symmetry. Our analysis of Table 7 indicates that the asymmetry introduced by HeNCler significantly improves performance on directed datasets. Specifically, we observe a noticeable gap between the performance of the 'Undirected' baseline and HeNCler, which supports the effectiveness of our design for directed graphs.
>
> For undirected graphs, the performance improvements are less pronounced. This aligns with our hypothesis that latent asymmetries are less critical for these datasets. However, HeNCler still achieves slight performance gains for undirected graphs, suggesting that it effectively captures subtle asymmetries that may exist even in nominally undirected data.
>
> The larger performance gains in directed graphs stem from HeNCler’s ability to simultaneously relax the homophily and symmetry assumptions. In contrast, the symmetry assumption is naturally satisfied in undirected datasets, limiting the potential impact of HeNCler’s asymmetry.
>
> **Q2: Low NMI Scores and Class Imbalance**
>
> We choose to use supervised metrics such as F1 and NMI, as they are commonly used in the literature. However, the assumption that good clustering should align with the given labels may not always hold. Determining the most suitable metric for clustering remains an open problem and is beyond the scope of this paper. Nevertheless, we observe that NMI tends to be particularly low for datasets with significant class imbalance. Despite these challenges, the relative improvements achieved by HeNCler demonstrate its efficacy under such conditions

---

> > ### Comment · Reviewer_8wDQ · 2024-11-27
> >
> > Thanks for the detailed analysis, which addressed my questions about Low NMI Scores and its relationship to class imbalance. The performance regarding direct and undirected graphs still requires further analysis. I have gone through others comments. And I finally decide to retain my original score.

---

> ### Comment · Area_Chair_UMwD · 2024-11-25
>
> Could please acknowledge and respond to the rebuttal.

---

### Official Review · Reviewer_9rSZ · 2024-11-05

**Soundness:** 3
**Presentation:** 2
**Contribution:** 2
**Rating:** 3
**Confidence:** 4

**Summary:**

This paper proposes a kernel spectral biclustering framework to learn an induced asymmetric similarity graph suited for node clustering of heterophilous graphs.

**Strengths:**

1. The proposed method is easy to understand.
2. It addresses an important problem.

**Weaknesses:**

1. The innovation seems limited, as graph rewiring is already a common approach to tackle heterophilous graphs, as seen in [1] and [2]. However, it would be beneficial for the authors to clarify how their kernel spectral biclustering approach meaningfully differs from existing graph rewiring techniques. Specifically, a comparison of how these methods address heterophily and clustering effectiveness would strengthen the argument for the novelty of their approach. By expanding on the specific advantages or unique contributions of kernel spectral biclustering over rewiring methods, the authors could better situate their work within the current landscape.

2. While the proposed method achieves the best performance in 11 out of 16 cases, the choice of only five baselines may limit the comprehensiveness of this evaluation. I suggest including additional baselines that are well-regarded in heterophilous graph clustering to provide a more robust comparison, like [4][5]. Additionally, reporting on the statistical significance of the observed improvements would help clarify whether these performance gains are practically meaningful or consistent across datasets.

3. More detailed analysis of Table 4 would improve the discussion of the results. Specifically, it would be helpful for the authors to discuss the relative impact of the different components (e.g., the kernel spectral biclustering loss vs. reconstruction losses) in achieving the overall performance. Additionally, an investigation into any observed trends or dependencies among these components would provide insights into the roles they play in model performance.

4. I recommend adding larger datasets, such as Ogbn-arxiv [3], to further validate the proposed method. Testing on larger datasets could help assess the scalability and robustness of the approach and reveal any computational challenges that might arise when handling high-dimensional data. This addition could provide a more comprehensive evaluation and help demonstrate the method's potential for broader applications.

[1] Jiang, W., Gao, X., Xu, G., Chen, T., & Yin, H. (2024, May). Challenging Low Homophily in Social Recommendation. In Proceedings of the ACM on Web Conference 2024 (pp. 3476-3484).
[2] Zheng, Y., Zhang, H., Lee, V., Zheng, Y., Wang, X., & Pan, S. (2023, July). Finding the missing-half: Graph complementary learning for homophily-prone and heterophily-prone graphs. In International Conference on Machine Learning (pp. 42492-42505). PMLR.
[3] Hu, W., Fey, M., Zitnik, M., Dong, Y., Ren, H., Liu, B., ... & Leskovec, J. (2020). Open graph benchmark: Datasets for machine learning on graphs. Advances in neural information processing systems, 33, 22118-22133.
[4] Li W Z, Wang C D, Xiong H, et al. Homogcl: Rethinking homophily in graph contrastive learning[C]//Proceedings of the 29th ACM SIGKDD Conference on Knowledge Discovery and Data Mining. 2023: 1341-1352.
[5] Gu M, Yang G, Zhou S, et al. Homophily-enhanced structure learning for graph clustering[C]//Proceedings of the 32nd ACM International Conference on Information and Knowledge Management. 2023: 577-586.

**Questions:**

See Weakness.

---

> ### Author Response · Authors · 2024-11-25
>
> **Weakness 1**
>
> Our literature review focuses on unsupervised clustering and representation learning techniques specifically designed for heterophilous graphs. While graph rewiring techniques, such as those referenced (e.g., [2]), are relevant, they are predominantly introduced in a supervised setting. As our method targets an unsupervised scenario, it diverges from these approaches in both objectives and methodology. We will clarify this distinction in our final paper.
>
> **Weakness 2**
>
> We agree that methods like [4] and [5] are valuable references; however, these approaches are designed to enhance homophily in already homophilous graphs, which contrasts with our aim of effectively addressing heterophilous graphs. Nonetheless, we will incorporate these related works into our literature review to provide a comprehensive view of the landscape and clearly articulate how our approach differs in terms of both goals and methodology.
>
> **Weakness 1 + 2**
>
> We appreciate your suggestion and will integrate a more extensive discussion of these related works into our literature review. Additionally, we will expand our baseline comparisons in the final version of the paper to include the most relevant and best performing methods, ensuring a robust evaluation of our approach.
>
> **Weakness 3**
>
> We appreciate the request for further elaboration on our ablation results. As shown in Tables 4 and 7, all loss components contribute positively to HeNCler's overall performance. For directed datasets, model asymmetry emerges as a significant contributor, while for undirected datasets, the reconstruction loss, asymmetry, and wKSVD loss contribute roughly equally. We will provide a more detailed discussion of these findings in the final version to offer deeper insights into the contributions of individual components.
>
> **Weakness 4**
>
> We understand the importance of evaluating our method on larger datasets. Unfortunately, as noted, there are currently no large-scale datasets in the literature that are intrinsically heterophilous. The ogbn-arxiv dataset is inherently homophilous, and the arxiv-year dataset (Lim et al., NeurIPS 2021) simply reassigns labels based on publication years, which lacks interpretability in an unsupervised setting. Nonetheless, we will include experiments on the arxiv-year dataset to assess the computational scalability of our method. These results will provide insights into the computational challenges and scalability of HeNCler, even if the heterophily aspect is not fully represented.

---

### Meta-Review · Area_Chair_UMwD · 2024-12-19

**Metareview:**

Clustering nodes in graphs where relationships are not symmetric (heterophilous graphs) is challenging for traditional methods that rely on high similarity within clusters. The proposed method, uses a technique to capture asymmetric relationships.

Strengths:

See comments by the reviewers.

Weaknesses:

1) limited innovation
2) More comparisons to baselines are needed
3) More comparisons to larger datasets
4) More analysis on comparisons to undirected graphs.
5) Better literature review.

The rebuttal didn't seem to address the concerns of the reviewers.
Overall, the paper needs major revision and cannot be accepted at this stage.

**Additional Comments On Reviewer Discussion:**

The reviewers raised a lot of concerns related to

1) limited innovation
2) More comparisons to baselines are needed
3) More comparisons to larger datasets
4) More analysis on comparisons to undirected graphs.
5) Better literature review.

The rebuttal didn't seem to address the concerns of the reviewers.
Overall, the paper needs major revision and cannot be accepted at this stage.

---

### Decision · Program_Chairs · 2025-01-22

Reject